# DIVERSE EXPLORATION VIA INFOMAX OPTIONS

## ABSTRACT

In this paper, we study the problem of autonomously discovering temporally abstracted actions, or options, for exploration in reinforcement learning. For learning diverse options suitable for exploration, we introduce the *infomax* termination objective defined as the mutual information between options and their corresponding state transitions. We derive a scalable optimization scheme for maximizing this objective via the termination condition of options, yielding the InfoMax Option Critic (IMOC) algorithm. Through illustrative experiments, we empirically show that IMOC learns diverse options and utilizes them for exploration. Moreover, we show that IMOC scales well to continuous control tasks.

## 1 INTRODUCTION

Abstracting a course of action as a higher-level action, or an *option* (Sutton et al., 1999), is a key ability for reinforcement learning (RL) agents in several aspects, including *exploration*. In RL problems, an agent learns to approximate an optimal policy only from experience, given no prior knowledge. This leads to the necessity of exploration: an agent needs to explore the poorly known states for collecting environmental information, sometimes sacrificing immediate rewards. For statistical efficiency, it is important to explore the state space in a *deep* and directed manner, rather than taking uniformly random actions (Osband et al., 2019). Options can represent such directed behaviors by capturing long state jumps from their starting regions to terminating regions. It has been shown that well-defined options can facilitate exploration by exploiting an environmental structure (Barto et al., 2013) or, more generally, by reducing decision steps (Fruit and Lazaric, 2017).

A key requirement for such explorative options is *diversity*. If all options have the same terminating region, they will never encourage exploration. Instead, options should lead to a variety of regions for encouraging exploration. However, automatically discovering diverse options in a scalable, online manner is challenging due to two difficulties: generalization and data limitation. Generalization with function approximation (Sutton, 1995) is important for scaling up RL methods to large or continuous domains. However, many existing option discovery methods for exploration are graph-based (e.g., Machado et al. (2017)) and incompatible with function approximation, except for that by Jinnai et al. (2020). Discovering options online in parallel with polices requires us to work with limited data sampled from the environment and train the model for evaluating the diversity in a data-efficient manner.

To address these difficulties, we introduce the *infomax* termination objective defined as the mutual information (MI) between options and their corresponding state transitions. This formulation reflects a simple inductive bias: for encouraging exploration, options should terminate in a variety of regions per starting regions. Thanks to the information-theoretical formulation, this objective is compatible with function approximation and scales up to continuous domains. A key technical contribution of this paper is the optimization scheme for maximizing this objective. Specifically, we employ a simple classification model over options as a *critic* for termination conditions, which makes our method data-efficient and tractable in many domains.

The paper is organized as follows. After introducing background and notations, we present the infomax termination objective and derive a practical optimization scheme using the termination gradient theorem (Harutyunyan et al., 2019). We then implement the infomax objective on the option-critic architecture (OC) (Bacon et al., 2017) with algorithmic modifications, yielding the InfoMax Option Critic (IMOC) algorithm. Empirically, we show that (i) IMOC improves exploration in structured environments, (ii) IMOC improves exporation in lifelong learning, (iii) IMOC is scalable

to MuJoCo continuous control tasks, and (iv) the options learned by IMOC are diverse and meaningful. We then relate our method to other option-learning methods and the *empowerment* concept (Klyubin et al., 2005), and finally give concluding remarks.

## 2 BACKGROUND AND NOTATION

We assume the standard RL setting in the Markov decision process (MDP), following Sutton and Barto (2018). An MDP $\mathcal{M}$ consists of a tuple $(\mathcal{X}, \mathcal{A}, p, r, \gamma)$, where $\mathcal{X}$ is the set of states, $\mathcal{A}$ is the set of actions, $p : \mathcal{X} \times \mathcal{A} \times \mathcal{X} \to [0, 1]$ is the state transition function, $r : \mathcal{X} \times \mathcal{A} \to [r_{\min}, r_{\max}]$ is the reward function, and $0 \leq \gamma \leq 1$ is the discount factor. A policy is a probability distribution over actions conditioned on a state $x$, $\pi : \mathcal{X} \times \mathcal{A} \to [0, 1]$. For simplicity, we consider the episodic setting where each episode ends when a terminal state $x_T$ is reached. In this setting, the goal of an RL agent is to approximate a policy that maximizes the expected discounted cumulative reward per episode:

$$J^{\mathrm{RL}}(\pi) = \mathbb{E}_{\pi, x_0} \left[ \sum_{t=0}^{T-1} \gamma^t R_t \right], \tag{1}$$

where $R_t = r(x_t, a_t)$ is the reward received at time $t$, and $x_0$ is the initial state of the episode. Relatedly, we define the action-value function $Q^\pi(x_t, a_t) \stackrel{\text{def}}{=} \mathbb{E}_{x_t, a_t, \pi} \left[ \sum_{t'=t}^{T-1} \gamma^{t'-t} R_{t'} \right]$ and the state-value function $V^\pi(x_t) \stackrel{\text{def}}{=} \mathbb{E}_{x_t, \pi} \sum_a \pi(a|x_t) Q^\pi(x_t, a)$.

Assuming that $\pi$ is differentiable by the policy parameters $\theta_\pi$, a simple way to maximize the objective (1) is the policy gradient method (Williams, 1992) that estimates the gradient by:

$$\nabla_{\theta_\pi} J^{\mathrm{RL}}(\pi) = \mathbb{E}_{\pi, x_t} \left[ \nabla_{\theta_\pi} \log \pi(a_t|x_t) \hat{A}(x_t, a_t) \right], \tag{2}$$

where $\hat{A}(x_t, a_t)$ is the estimation of the advantage function $A^\pi(x_t, a_t) \stackrel{\text{def}}{=} Q^\pi(x_t, a_t) - V^\pi(x_t)$. A common choice of $\hat{A}(x_t, a_t)$ is $N$-step TD error $\sum_{i=0}^{N} \gamma^i R_{t+i} + \gamma^N \hat{V}(x_{t+N}) - \hat{V}(x_t)$, where $N$ is a fixed rollout length (Mnih et al., 2016).

### 2.1 OPTIONS FRAMEWORK

Options (Sutton et al., 1999) provide a framework for representing temporally abstracted actions in RL. An option $o \in \mathcal{O}$ consists of a tuple $(\mathcal{I}^o, \beta^o, \pi^o)$, where $\mathcal{I}^o \subseteq \mathcal{X}$ is the initiation set, $\beta^o : \mathcal{X} \to [0, 1]$ is a termination function with $\beta^o(x)$ denoting the probability that option $o$ terminates in state $x$, and $\pi^o$ is *intra-option* policy. Following related studies (Bacon et al., 2017; Harutyunyan et al., 2019), we assume that $\mathcal{I}^o = \mathcal{X}$ and learn only $\beta^o$ and $\pi^o$. Letting $x_s$ denote an option-starting state and $x_f$ denote an option-terminating state, we can write the option transition function as:

$$P^o(x_f|x_s) = \beta^o(x_f) \mathbb{I}_{x_f = x_s} + (1 - \beta^o(x_s)) \sum_x p^{\pi^o}(x|x_s) P^o(x_f|x), \tag{3}$$

where $\mathbb{I}$ is the indicator function and $p^{\pi^o}$ is the *policy-induced* transition function $p^{\pi^o}(x'|x) \stackrel{\text{def}}{=} \sum_{a \in \mathcal{A}} \pi^o(a|x) p(x'|x, a)$. We assume that all options eventually terminate so that $P^o$ is a valid probability distribution over $x_f$, following Harutyunyan et al. (2019).

To present option-learning methods, we define two option-value functions: $Q_\mathcal{O}$ and $U_\mathcal{O}$, where $Q_\mathcal{O}$ is the option-value function denoting the value of selecting an option $o$ at a state $x_t$ defined by $Q_\mathcal{O}(x_t, o) \stackrel{\text{def}}{=} \mathbb{E}_{\pi, \beta, \mu} \left[ \sum_{t'=t}^{T-1} \gamma^{t'-t} R_t \right]$. Analogously to $Q^\pi$ and $V^\pi$, we let $V_\mathcal{O}$ denote the marginalized option-value function $V_\mathcal{O}(x) \stackrel{\text{def}}{=} \sum_o \mu(o|x) Q_\mathcal{O}(x, o)$, where $\mu(o|x_s) : \mathcal{X} \times \mathcal{O} \to [0, 1]$ is the policy over options. Function $U_\mathcal{O}(x, o) \stackrel{\text{def}}{=} (1 - \beta^o(x)) Q_\mathcal{O}(x, o) + \beta^o(x) V_\mathcal{O}(x)$ is called the option-value function *upon arrival* (Sutton et al., 1999) and denotes the value of reaching a state $x_t$ with $o$ and not having selected the new option.

### 2.2 OPTION CRITIC ARCHITECTURE

OC (Bacon et al., 2017) provides an end-to-end algorithm for learning $\pi^o$ and $\beta^o$ in parallel. To optimize $\pi^o$, OC uses the intra-option policy gradient method that is the option-conditional version

of the gradient estimator (2), $\nabla_{\theta_{\pi^o}} J^{\text{RL}}(\pi^o) = \mathbb{E}\big[\nabla_{\theta_{\pi^o}} \log \pi^o(a_t|x_t)\hat{A}^o(x_t, a_t)\big]$, where $\hat{A}^o$ is an estimation of the option-conditional advantage $A^{\pi^o}$.

For optimizing $\beta^o$, OC directly maximizes $Q_{\mathcal{O}}$ using the estimated gradient:

$$\nabla_{\theta_{\beta^o}} Q_{\mathcal{O}}(x, o) = \gamma \mathbb{E}\left[-\nabla_{\theta_{\beta^o}} \beta^o(x)\Big(Q_{\mathcal{O}}(x, o) - V_{\mathcal{O}}(x)\Big)\right]. \tag{4}$$

Intuitively, this decreases the termination probability $\beta^o(x)$ when holding an $o$ is advantageous, i.e., $Q_{\mathcal{O}}(x) - V_{\mathcal{O}}(x)$ is positive, and vice versa. Our method basically follows OC but has a different objective for learning $\beta^o$.

## 2.3 TERMINATION CRITIC

Recently proposed termination critic (TC) (Harutyunyan et al., 2019) optimizes $\beta^o$ by maximizing the information-theoretic objective called *predictability*:

$$J^{\text{TC}}(P^o) = -H(X_f|o), \tag{5}$$

where $H$ denotes entropy and $X_f$ is the random variable denoting the option-terminating states. Maximizing $-H(X_f|o)$ makes the terminating region of an option smaller and more predictable. In other words, we can *compress* terminating regions by optimizing the objective (5). To derivate this objective by the beta parameters $\theta_{\beta^o}$, Harutyunyan et al. (2019) introduced the termination gradient theorem:

**Theorem 1.** *Let $\beta^o$ be parameterized with a sigmoid function and $\ell_{\beta^o}$ denote the logit of $\beta^o$. We have*

$$\nabla_{\theta_\beta} P^o(x_f|x_s) = \sum_x P^o(x|x_s)\nabla_{\theta_\beta} \ell_{\beta^o}(x)(\mathbb{I}_{x_f=x} - P^o(x_f|x)), \tag{6}$$

Leveraging the theorem 1, TC performs gradient ascent using the estimated gradient:

$$\nabla_{\theta_{\beta^o}} J^{\text{TC}}(P^o) = -\mathbb{E}_{x_s,x,x_f}\left[\nabla_{\theta_\beta} \ell_{\beta^o}(x)\beta^o(x)\left(\Big(\log P^o_\mu(x) - \log P^o_\mu(x_f)\Big) + \Big(1 - \frac{P^o(x_f|x_s)P^o_\mu(x)}{P^o_\mu(x_f)P^o(x|x_s)}\Big)\right)\right].$$

where $P^o_\mu(x)$ is the marginalized distribution of option-terminating states.

Contrary to the termination objective of OC (4), this objective does not depend on state values, making learned options robust against the reward structure of the environment. Our method is inspired by TC and optimizes a similar information-theoretic objective, not for predictability but for *diversity*. Also, our infomax objective requires an estimation of $\hat{p}(o|x_s, x_f)$ instead of the option transition model $P^o(x_f|x_s)$, which makes our method tractable in more environments.

## 3 INFOMAX OPTION CRITIC

We now present the key idea behind the InfoMax Option Critic (IMOC) algorithm. We first formulate the infomax termination objective based on the MI maximization, then derive a practical gradient estimation for maximizing this objective on $\beta^o$, utilizing the termination gradient theorem 1.

To evaluate the diversity of options, we use the MI between options and option-terminating states conditioned by option-starting states:

$$J^{\text{IMOC}} = I(X_f; O|X_s) = H(X_f|X_s) - H(X_f|X_s, O), \tag{7}$$

where $I$ denotes conditional MI $I(A; B|Z) = H(A|Z) - H(A|B, Z)$, $X_s$ is the random variable denoting an option-starting state, and $O$ is the random variable denoting an option. We call this objective the *infomax* termination objective. Let us interpret $X_f|X_s$ as the random variable denoting a state transition induced by an option. Then maximizing the MI (7) (i) diversifies a state transition $X_f|X_s$ and (ii) makes an option-conditional state transition $X_f|X_s, o$ more deterministic. Note that the marginalized MI $I(X_f; O)$ also makes sense in that it prevents the terminating region of each option from being too broad, as predictability (5) does. However, in this study, we focus on the conditional objective since it is easier to optimize.

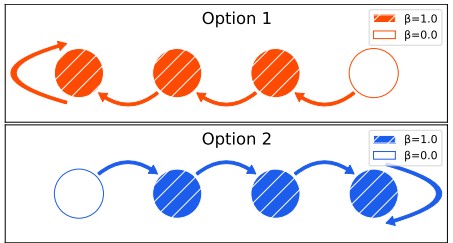 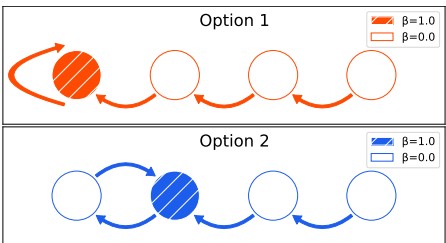

Figure 1: Two insances of InfoMax options in the four state deterministic chain. **Left:** Options are diverse but all state transitions per option are one-step. **Right:** Options enable relatively long state transitions but option-policies are the same at some states.

To illustrate the limitation of infomax options, we conducted an analysis in a toy four-state *deterministic chain* environment, which has four states and two deterministic actions (go left and go right) per each state. Since deriving the exact solution is computationally difficult, we searched options that maximize $H(X_f|X_s)$ from deterministic options that has deterministic option-policies and termination functions (thus has the minimum $H(X_f|X_s, O)$). Among multiple solutions, Figure 1 shows two interesting instances of deterministic infomax options when $|\mathcal{O}| = 2$. The left options enable diverse behaviors per state, although they fail to capture long-term behaviors generally favorable in the literature (e.g., Mann et al. (2015)). On the other hand, the right options enable relatively long, two step state transitions, but they are the same and the rightmost state and the next one. Furthermore, an agent can be caught in a small loop that consists of the leftmost state and the next one. This example shows that (i) we can obtain short and diverse options with only a few options, (ii) to obtain long and diverse options, we need sufficiently many options, and (iii) an agent can be caught in a small loop with only a few options, failing to visit diverse states. As we show in Appendix A, this 'small loop' problem cannot happen with four options. Thus, the number of options is important when we are to maximize the MI (7) and a limitation of this method. However, in experiments, we show that we can practically learn diverse options with relatively small number of options.

For maximizing the MI by gradient ascent, we now derive the gradient of the infomax termination objective (7). First, we estimate the gradient of the objective using the option transition model $P^o$ and marginalized option-transition model $P(x_f|x_s) = \sum_o \mu(o|x_s)P^o(x_f|x_s)$.

**Proposition 1.** *Let $\beta^o$ be parameterized with a sigmoid function. Given a trajectory $\tau = x_s, \ldots, x, \ldots, x_f$ sampled by $\pi^o$ and $\beta^o$, we can obtain unbiased estimations of $\nabla_{\theta_\beta} H(X_f|X_s)$ and $\nabla_{\theta_\beta} H(X_f|X_s, O)$ by*

$$\nabla_{\theta_\beta} H(X_f|X_s) = \mathbb{E}_{x_s, x, x_f, o} \left[ -\nabla_{\theta_\beta} \ell_{\beta^o}(x)\beta^o(x) \Big( \log P(x|x_s) - \log P(x_f|x_s) \Big) \right] \tag{8}$$

$$\nabla_{\theta_\beta} H(X_f|X_s, O) = \mathbb{E}_{x_s, x, x_f, o} \left[ -\nabla_{\theta_\beta} \ell_{\beta^o}(x)\beta^o(x) \Big( \log P^o(x|x_s) - \log P^o(x_f|x_s) \Big) \right] \tag{9}$$

*where $\ell_{\beta^o}(x)$ denotes the logit of $\beta^o(x)$.*

Note that the additional term $\beta^o$ is necessary because $x$ is not actually a terminating state. The proof follows section 4 in Harutyunyan et al. (2019) and is given in Appendix B.1.

The estimated gradient of the infomax termination objective (7) can now be written as:

$$\nabla_{\theta_\beta} I(X_f; O|X_s) = \nabla_{\theta_\beta} H(X_f|X_s) - \nabla_{\theta_\beta} H(X_f|X_s, O)$$
$$= \mathbb{E}_{x_s, x, x_f, o} \left[ -\nabla_{\theta_\beta} \ell_{\beta^o}(x)\beta^o(x) \left( \log P(x|x_s) - \log P(x_f|x_s) - (\log P^o(x|x_s) - \log P^o(x_f|x_s)) \right) \right],$$
$$\tag{10}$$

which means that we can optimize this objective by estimating $P^o$ and $P$. However, estimating the probability over the state space can be difficult, especially when the state space is large, as common in the deep RL setting. Hence, we reformulate the gradient using Bayes' rule in a similar way as Gregor et al. (2017). The resulting term consists of the reverse option transition $p(o|x_s, x_f)$ that denotes the probability of having an $o$ given a state transition $x_s, x_f$.

**Proposition 2.** *We now have*

$$\nabla_{\theta_\beta} I(X_f; O|X_s) = \nabla_{\theta_\beta} H(X_f|X_s) - \nabla_{\theta_\beta} H(X_f|X_s, O)$$
$$= \mathbb{E}_{x_s, x, x_f, o}\left[\nabla_{\theta_\beta} \ell_{\beta^o}(x)\beta^o(x)\Big(\log p(o|x_s, x) - \log p(o|x_s, x_f)\Big)\right] \quad (11)$$

The proof is given in Appendix B.2. In the following sections, we estimate the gradient (11) by learning a classification model over options $\hat{p}(o|x_s, x_f)$ from sampled option transitions.

## 4 ALGORITHM

In this section, we introduce modifications for adjusting the OC (Bacon et al., 2017) to our infomax termination objective. Specifically, we implement IMOC on top of Advantage-Option Critic (AOC), a synchronous variant of A2OC (Harb et al., 2018), yielding the Advantage-Actor InfoMax Option Critic (A2IMOC) algorithm. To stably estimates $\hat{p}(o|x_s, x_f)$ for updating $\theta_\beta$, we sample recent option state transitions $o, x_s, x_f$ from We follow AOC for optimizing option-policies except the following modifications and give a full description of A2IMOC in Appendix C.1. In continuous control experiments, we also used Proximal Policy InfoMax Option Critic (PPIMOC) that is an implementation of IMOC based on PPO (Schulman et al., 2017). We give details of PPIMOC in Appendix C.2.

**Upgoing Option-Advantage Estimation**  Previous studies (e.g., Harb et al. (2018)) estimated the advantage $\hat{A}^{o_t}(x_t)$ ignoring the future rewards after the current option $o_t$ terminates. Since longer rollout length often helps speed up learning (Sutton and Barto, 2018), it is preferable to extend this estimation to use all available future rewards. However, future rewards after option termination heavily depends on the selected option, often leading to underestimation of $\hat{A}^o$. Thus, to effectively use future rewards, we introduce an *upgoing* option-advantage estimation (UOAE). Let $t + k$ denote the time step where the current option $o_t$ terminates in a sampled trajectory. Then, UOAE estimates the advantage by:

$$\hat{A}^o_{\text{UOAE}} = -Q_{\mathcal{O}}(x_t, o_t) + \begin{cases} \sum_{i=0}^{k} \gamma^i R_{t+i} + \underbrace{\max\left(\sum_{j=k}^{N} \gamma^j R_{t+j}, \ \gamma^k V_{\mathcal{O}}(x_{t+k})\right)}_{\text{upgoing estimation}} & (k < N) \\ \sum_{i=0}^{N} \gamma^i R_{t+i} + \gamma^N U_{\mathcal{O}}(x_{t+N}, o_t) & (otherwise) \end{cases}.$$
$$(12)$$

Similar to upgoing policy update (Vinyals et al., 2019), the idea is to be optimistic about the future rewards after option termination by taking the maximum with $V_{\mathcal{O}}$.

**Policy Regularization Based on Mutual Information**  To perform MI maximization not only on termination functions but also on option-policies, we introduce a policy regularization based on the maximization of the conditional MI, $I(A; O|X_s)$, where $A$ is the random variable denoting an action. This MI can be interpreted as a local approximation of the infomax objective (7), assuming that each action leads to different terminating regions. Although optimizing the infomax termination objective diversifies option-policies implicitly, we found that this regularization helps learn diverse option-policies reliably. Letting $\pi_\mu$ denote the marginalized policy $\pi_\mu(a|x) \stackrel{\text{def}}{=} \sum_o \mu(o|x)\pi^o(a|x)$, we write $I(A; O|X_s)$ as:

$$I(A; O|X_s) = H(A|X_s) - H(A|O, X_s) = \mathbb{E}_{x_s}[H(\pi_\mu(x_s))] - \mathbb{E}_{x_s, o}[H(\pi^o(x_s))].$$

We use this regularization with the entropy bonus (maximization of $H(\pi^o)$) common in policy gradient methods (Williams and Peng, 1991; Mnih et al., 2016) and write the overall regularization term as

$$c_{H_\mu} H(\pi_\mu(x)) + c_H H(\pi^o(x)),$$
$$(13)$$

where $c_{H_\mu}$ and $c_H$ are weights of each regularization term. Note that we add this regularization term on not only option-starting states but all sampled states. This introduces some bias, which we did

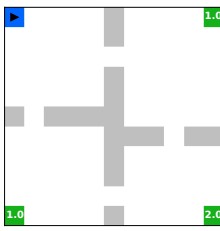

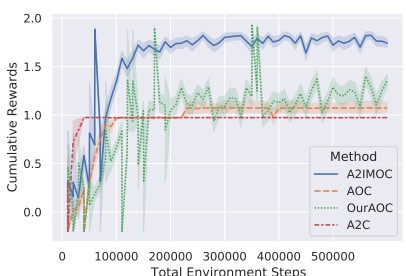

(a) Gridworld environment.
Blue grid is the start and green grids are goals.

(b) Performance progression.

Figure 2: Single Task learning in Gridworld Four Rooms.

not find to be harmful when $c_{H_\mu}$ is reasonably small. To approximate $H(\pi_\mu)$, we employ $\hat\mu$ that is the empirical estimation of $\mu$. Using $\hat\mu$, $H(\pi_\mu)$ is computed by $\pi_\mu(a|x) \approx \sum_o \hat\mu(o|x)\pi^o(a|x)$ for descrete action spaces and approximated by Monte Carlo method for continuous action spaces. We show the details in Appendix C.1.

## 5 EXPERIMENTS

We conducted a series of experiments to show two use cases of IMOC: exploration in structured environments and exploration for lifelong learning (Brunskill and Li, 2014). In this section, we used four options for all option-learning methods and compared the number of options in Appendix D.7.

### 5.1 SINGLE TASK LEARNING IN STRUCTURED ENVIRONMENTS

We consider two 'Four Rooms' domains, where diverse options are beneficial for utilizing environmental structures.

**Gridworld Four Rooms with Suboptimal Goals** First, we tested IMOC in a variant of the classical Four Rooms Gridworld (Sutton et al., 1999) with suboptimal goals. An agent is initially placed at the upper left room and receives a positive reward only at goal states: two closer goals with $+1$ reward and the farthest goal with $+2$ reward, as shown in Figure 2a. The episode ends when an agent reaches one of the goals. The optimal policy is aiming the farthest goal in the lower right room without converging to suboptimal goals. Thus, an agent is required to learn multimodal behaviors leading to multiple goals, which options can help. In this environment, we compared A2IMOC with A2C (Mnih et al., 2016; Wu et al., 2017), AOC, and our tuned version of AOC (our AOC) with all enhancements presented in section 4 to highlight the effectiveness of the termination objective among all of our improvements.[1] We show the progress of average cumulative rewards over ten trials in Figure 2b. A2IMOC performed the best and found the optimal goal in most trials. AOC and our AOC also occasionally found the optimal goal, while A2C overfitted to either of the suboptimal goals through all trials.

Figure 3 illustrates learned option-polices and termination functions of each compared method.[2] Terminating regions learned with A2IMOC is diverse. For example, option 0 mainly terminates in the right rooms while option 3 terminates in the left rooms. We see that termination regions learned by A2IMOC are diverse and clearly separated per each option. Although all option policies converged to the same near optimal one, we show that A2IMOC diversifies option-policies at the beginning of learning in Appendix D.4. On the other hand, terminating regions learned with AOC overlap each other, and notably, option 3 has no terminating region. We assume this is because the loss function (4) decreases the terminating probability when the advantage is positive. We can see the same tendency in our AOC, although we cannot see the vanishment of the terminating regions.

---

[1] Note that we did not include ACTC (Harutyunyan et al., 2019) for comparison since we failed to reliably reproduce the reported results with our implementation of ACTC.

[2] Note that we choose the best model from multiple trials for visualization throughout the paper.

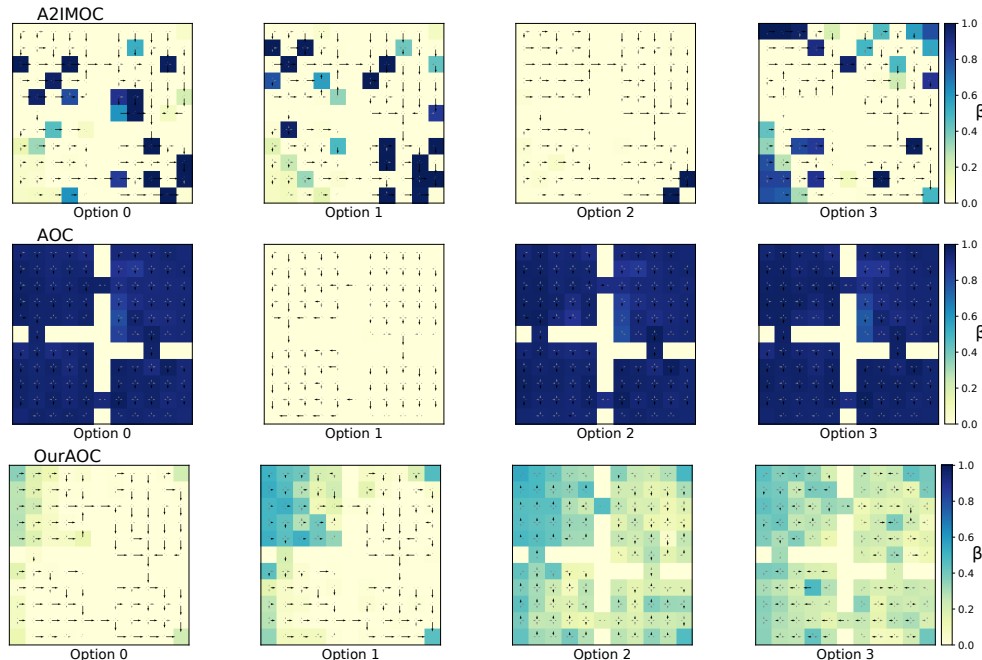

Figure 3: Learned option-policies ($\pi^o$) and termination probabilities ($\beta^o$) for each option in Gridworld Four Rooms after $6 \times 10^5$ steps. Arrows show the probablities of each action and heatmaps show probabilities of each $\beta^o$. **First row:** A2IMOC. Terminating regions are clearly different each other and there are a few overlapped regions. **Second row:** AOC. Almost all states has high termination probablity with option $0, 1, 3$ and option 2 has no terminating region. **Third row:** Our AOC. Termination regions are not clearly separated per each option.

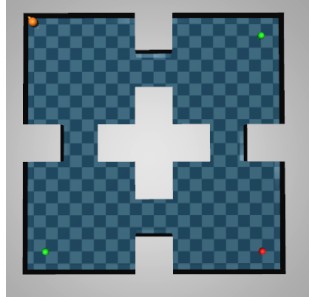

(a) MuJoCo Point Four Rooms. The agent is an orange ball and there are three goals: green goals are suboptimal ($+0.5$) and the red one in optimal ($+1.0$).

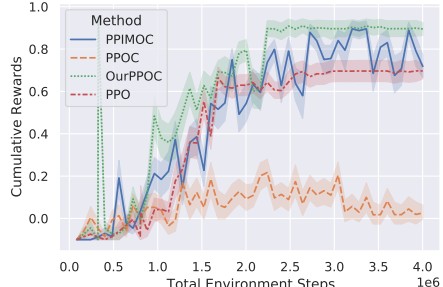

(b) Performance progression.

Figure 4: Single Task learning in MuJoCo Point Four Rooms.

**MuJoCo Point Four Rooms** To show the scalability of IMOC in continuous domains, we conducted experiments in a similar four rooms environment, based on the MuJoCo (Todorov et al., 2012) physics simulator and "PointMaze" environment in rllab (Duan et al., 2016). This environment follows the Gridworld Four Rooms and has three goals as shown in Figure 4a: two green goals with $+0.5$ reward, and a red one with $+1.0$. An agent controls the rotation and velocity of the orange ball and receives a positive reward only at the goals. In this environment, we compared PPIMOC with PPOC (Klissarov et al., 2017), PPO, and our tuned version of PPOC (our PPOC) that is the same as PPIMOC except for the termination objective. An important difference is that PPOC uses a parameterized $\mu$ trained by policy gradient, while PPIMOC and our PPOC use $\epsilon$-greedy for option selection. Figure 4b show the progress of average cumulative rewards over five trials. PPIMOC found the optimal goal four times in five trials and performed slightly bettern than PPO. In a qualitative

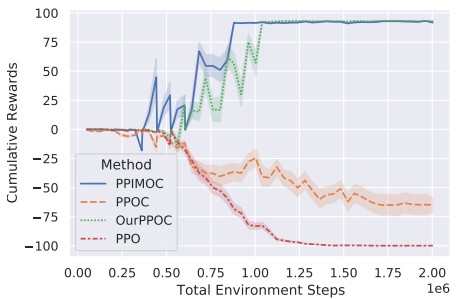

(a) Performance progression in Mountain Car.

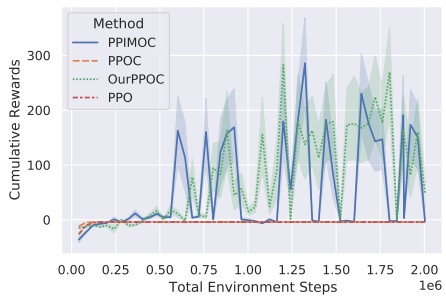

(b) Performance progression in Cartpole swing up.

Figure 5: Single Task learning in MuJoCo Point Four Rooms.



(a) MuJoCo Point Billiard. Four goals periodically move clockwise.

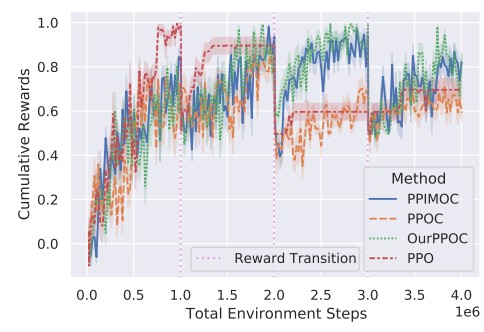

(b) Performance progression.

Figure 6: Lifelong learning in MuJoCo Point Billiard.

analysis, we observed the same tendency in learned options as the Gridworld experiment, where the details are given in Appendix D.5.

## 5.2 SINGLE TASK LEARNING IN CLASSICAL CONTINUOUS CONTROL

Additionally, we test PPIMOC on two classical, hard-exploration control problems: Mountain Car (Barto et al., 1983) and Cartpole swingup (Moore, 1990). In Mountain Car, PPIMOC and our PPOC successfully learns to reache the goal, while PPO and PPOC converged to run around the start posisition. In Cartpole swing up, PPIMOC and our PPOC performed better than PPO and PPOC, but still failed to learn stable behaviors.

## 5.3 EXPLORATION FOR LIFELONG LEARNING

As another interesting application of IMOC, we consider the lifelong learning setting. Specifically, we tested IMOC in 'Point Billiard' environment. In this environment, an agent receives a positive reward only when the blue *objective* ball reaches the goal, pushed by the agent (orange ball). Figure 6a shows all four configurations of Point Billiard that we used. There are four goals: green goals with $+0.5$ reward and a red one with $+1.0$ reward. The positions of four goals move clockwise after 1M environmental steps and agents need to adapt to the new positions of goals. We compared PPIMOC with PPO, PPOC, and our PPOC in this environment. Figure 6b shows the progress of average cumulative rewards over five trials. Both PPIMOC and our PPOC performed the best and adapted to all reward transitions. On the other hand, PPO and PPOC struggle to adapt to the second transition, where the optimal goal moves behind the agent. The ablation study given in Appendix D.6 shows that UOAE (12) works effectively in this task. However, without UOAE, PPIMOC still outperformed PPO. Thus, we argue that having diverse terminating regions itself is beneficial for adapting to new reward functions in environments with subgoals.

## 6 RELATED WORK

**Options for Exploration**    Options (Sutton et al., 1999) in RL are widely studied for many applications, including speeding up planning (Mann and Mannor, 2014) and transferring skills (Konidaris and Barto, 2007; Castro and Precup, 2010). However, as discussed by Barto et al. (2013), their benefits for exploration are less well recognized. Many existing methods focused on discovering subgoals that effectively decompose the problem then use such subgoals for encouraging exploration. Subgoals are discovered based on various properties, including graphical features of state transitions (Simsek and Barto, 2008; Machado et al., 2017; Jinnai et al., 2019) and causality (Jonsson and Barto, 2006; Vigorito and Barto, 2010). In contrast, our method directly optimizes termination functions instead of discovering subgoals, capturing environmental structures implicitly. From a theoretical perspective, Fruit and Lazaric (2017) analyzed that good options can improve the exploration combined with state-action visitation bonuses. Using infomax options with visitation bonuses would be an interesting future direction.

**End-to-end learning of Options**    While many studies attempted to learn options and option-policies separately, Bacon et al. (2017) proposed OC to train option-policies and termination functions in parallel.  OC has been extended with various types of inductive biases, including deliberation cost (Harb et al., 2018), interest (Khetarpal et al., 2020), and safety (Jain et al., 2018). Our study is directly inspired by an information-theoretic approach presented by Harutyunyan et al. (2019), as we noted in section 2.

**Mutual Information and Skill Learning**    MI often appears in the literature of *intrinsically motivated* (Singh et al., 2004) reinforcement learning, as a driver of goal-directed behaviors. A well-known example is the *empowerment* (Klyubin et al., 2005; Salge et al., 2013), which is obtained by maximizing the MI between sequential $k$ actions and the resulting state $I(a_t, ..., a_{t+k}; x_{t+k}|x_t)$. Some works (Mohamed and Rezende, 2015; Zhao et al., 2020) implemented lower bound maximization of empowerment as intrinsic rewards for RL agents, encouraging goal-directed behaviors in the absence of extrinsic rewards We can interpret our objective $I(X_f; O|X_s)$ as empowerment between limited action sequences and states corresponding to options. Gregor et al. (2017) employed this interpretation and introduced a method for maximizing the variational lower bound of this MI via option-policies, using the same model as our $\hat{p}$, while we aim to maximize the MI via termination functions. MI is also used for intrinsically motivated discovery of skills, assuming that diversity is important to acquire useful skills. Eysenbach et al. (2019) proposed to maximize MI between skills and states $I(O; X)$, extended to the conditional one $I(O; X'|X)$ by Sharma et al. (2020). Although our study shares the same motivation for using MI as these methods, i.e. diversifying sub-policies, the process of MI maximization is significantly different: our method optimizes termination functions, while their methods optimize conditional policies by using MI as intrinsic rewards.

## 7 CONCLUSION

We presented a novel end-to-end option learning algorithm InfoMax Option Critic (IMOC) that uses the infomax termination objective to diversify options. Empirically, we showed that IMOC improves exploration in structured environments and for lifelong learning, even in continuous control tasks. We also quantitatively showed the diversity of learned options. An interesting future direction would be combining our method for learning termination conditions with other methods for learning option-policies, e.g., by using MI as intrinsic rewards. A limitation of the infomax objective presented in this study is that it requires *on-policy* data for training. Hence, another interesting line of future work is extending IMOC to use for *off-policy* option discovery.

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

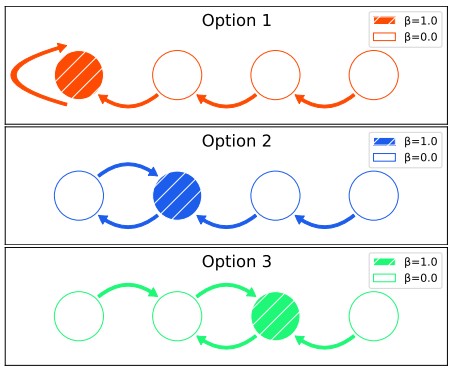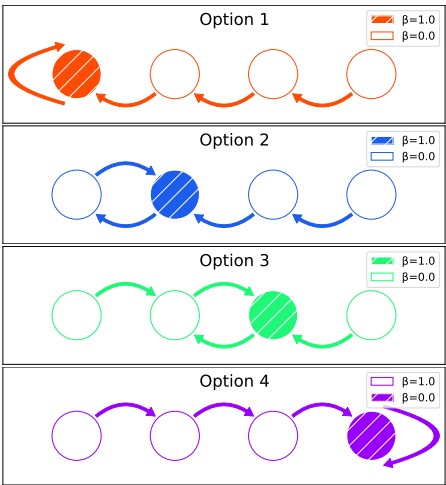

Figure 7: InfoMax options in the four state deterministic chain. **Left:** With three options. **Right:** With four options.

## A    MORE ANALYSIS ON DETERMINISTIC CHAIN EXAMPLE

Figure 7 shows infomax options with three options and four options in the four state deterministic chain example. Among multiple solutions, we selected options with an absorbing state per option (i.e., $\beta^o(x) = 1.0$ for only one $x$), which are partially the same as the right options in Figure 1. With four options, $\Pr(x_f|x_s) = 0.25$ for all $x_f$ and $x_s$, thus $H(X_f|X_s)$ is the maximum. This example shows that we need sufficiently many options for maximizing the MI, otherwise an agent can be caught in a small loop as we described in Section 3.

## B    OMITTED PROOFS

### B.1    PROOF OF PROPOSITION 1

First, we repeat the assumption 2 in Harutyunyan et al. (2019).

**Assumption 1.** *The distribution $d^\mu(\cdot|o)$ over the starting states of an option $o$ under policy $\mu$ is independent of its termination condition $\beta^o$.*

Note that this assumption does not strictly hold since $\epsilon$-Greedy option selection depends on $\beta^o$ via $Q_{\mathcal{O}}$. However, since this dependency is not so strong, we found that $\beta^o$ reliably converged in our experiments.

**Lemma 1.** *Assume that the distribution $d^\mu(\cdot|o)$ over the starting state of an option $o$ under policy $\mu$ is independent with $\beta^o$. Then the following equations hold.*

$$\nabla_{\theta_\beta} H(X_f|X_s) = -\sum_{x_s,o} d^\mu(x_s,o) \sum_x P^o(x|x_s) \nabla_{\theta_\beta} \ell_{\beta^o}(x) \Big[ \log P(x|x_s) + 1 - \sum_{x_f} P^o(x_f|x) \Big( \log P(x_f|x_s) + 1 \Big) \Big]$$

(14)

$$\nabla_{\theta_\beta} H(X_f|X_s,O) = -\sum_{x_s,o} d^\mu(x_s,o) \sum_x P^o(x|x_s) \nabla_{\theta_\beta} \ell_{\beta^o}(x) \Big[ \log P^o(x|x_s) + 1 - \sum_{x_f} P^o(x_f|x) \Big( \log P^o(x_f|x_s) + 1 \Big) \Big],$$

(15)

Sampling $x_s, x, x_f, o$ from $d^\mu$ and $P^o$,

$$\nabla_{\theta_\beta} H(X_f|X_s) = \mathbb{E}_{x_s,x,x_f,o} \Big[ -\nabla_{\theta_\beta} \ell_{\beta^o}(x) \beta^o(x) \Big( \log P(x|x_s) - \log P(x_f|x_s) \Big) \Big]$$

(16)

$$\nabla_{\theta_\beta} H(X_f|X_s,O) = \mathbb{E}_{x_s,x,x_f,o} \Big[ -\nabla_{\theta_\beta} \ell_{\beta^o}(x) \beta^o(x) \Big( \log P^o(x|x_s) - \log P^o(x_f|x_s) \Big) \Big].$$

(17)

**Proof of Lemma 1**

*Proof.* First, we prove Equation (14). Let $d^\mu(x_s)$ denote the probability distribution over $x_s$ under the policy $\mu$, or the marginal distribution of $d^\mu(x_s|o)$, and $d^\mu(x_s, o)$ denote the joint distribution of $x_s$ and $o$. Then, we have:

$$\nabla_{\theta_\beta} H(X_f|X_s) = -\nabla_{\theta_\beta} \sum_{x_s} d^\mu(x_s) \sum_{x_f} P(x_f|x_s) \log P(x_f|x_s)$$

$$= -\sum_{x_s} d^\mu(x_s) \sum_{x_f} \left( \nabla_{\theta_\beta} P(x_f|x_s) \log P(x_f|x_s) + P(x_f|x_s) \frac{\nabla_{\theta_\beta} P(x_f|x_s)}{P(x_f|x_s)} \right)$$

$$= -\sum_{x_s} d^\mu(x_s) \sum_{x_f} \nabla_{\theta_\beta} P(x_f|x_s) \Big( \log P(x_f|x_s) + 1 \Big)$$

$$= -\sum_{x_s} d^\mu(x_s) \sum_{x_f} \sum_{o} \mu(o|x_s) \underbrace{\nabla_{\theta_\beta} P^o(x_f|x_s)}_{\text{Apply theorem (6)}} \Big( \log P(x_f|x_s) + 1 \Big)$$

$$= -\sum_{x_s} d^\mu(x_s) \sum_{x_f} \sum_{o} \mu(o|x_s) \sum_{x} P^o(x|x_s) \nabla_{\theta_\beta} \ell_{\beta^o}(x) (\mathbb{I}_{x_f=x} - P^o(x_f|x)) \Big( \log P(x_f|x_s) + 1 \Big)$$

$$= -\sum_{x_s} d^\mu(x_s) \sum_{o} \mu(o|x_s) \sum_{x} P^o(x|x_s) \nabla_{\theta_\beta} \ell_{\beta^o}(x) \sum_{x_f} (\mathbb{I}_{x_f=x} - P^o(x_f|x)) \Big( \log P(x_f|x_s) + 1 \Big)$$

$$= -\underbrace{\sum_{x_s,o} d^\mu(x_s, o)}_{\text{sample}} \underbrace{\sum_{x} P^o(x|x_s)}_{\text{sample}} \nabla_{\theta_\beta} \ell_{\beta^o}(x) \times \Big[ \log P(x|x_s) + 1 - \underbrace{\sum_{x_f} P^o(x_f|x)}_{\text{sample}} \Big( \log P(x_f|x_s) + 1 \Big) \Big].$$

Sampling $x_s, x, x_f, o$, we get (16).

Then we prove Equation (15).

$$\nabla_{\theta_\beta} H(X_f|X_s, O) = -\nabla_{\theta_\beta} \sum_{x_s,o} d^\mu(x_s, o) \sum_{x_f} P^o(x_f|x_s) \log P^o(x_f|x_s)$$

$$= -\sum_{x_s,o} d^\mu(x_s, o) \sum_{x_f} \left( \nabla_{\theta_\beta} P^o(x_f|x_s) \log P^o(x_f|x_s) + P^o(x_f|x_s) \frac{\nabla_{\theta_\beta} P^o(x_f|x_s)}{P^o(x_f|x_s)} \right)$$

$$= -\sum_{x_s,o} d^\mu(x_s, o) \sum_{x_f} \underbrace{\nabla_{\theta_\beta} P^o(x_f|x_s)}_{\text{Apply theorem (6)}} \Big( \log P^o(x_f|x_s) + 1 \Big)$$

$$= -\sum_{x_s,o} d^\mu(x_s, o) \sum_{x_f} \sum_{x} P^o(x|x_s) \nabla_{\theta_\beta} \ell_{\beta^o}(x) (\mathbb{I}_{x_f=x} - P^o(x_f|x)) \Big( \log P^o(x_f|x_s) + 1 \Big)$$

$$= -\sum_{x_s,o} d^\mu(x_s, o) \sum_{x} P^o(x|x_s) \nabla_{\theta_\beta} \ell_{\beta^o}(x) \sum_{x_f} (\mathbb{I}_{x_f=x} - P^o(x_f|x)) \Big( \log P^o(x_f|x_s) + 1 \Big)$$

$$= -\underbrace{\sum_{x_s,o} d^\mu(x_s, o)}_{\text{sample}} \underbrace{\sum_{x} P^o(x|x_s)}_{\text{sample}} \nabla_{\theta_\beta} \ell_{\beta^o}(x) \times \Big[ \log P^o(x|x_s) + 1 - \underbrace{\sum_{x_f} P^o(x_f|x)}_{\text{sample}} \Big( \log P^o(x_f|x_s) + 1 \Big) \Big]$$

$\square$

Sampling $x_s, x, x_f, o$, we get (17).

### B.2 PROOF OF PROPOSITION 2

*Proof.* First, we have that:

$$\log P^o(x_f|x_s) - \log P(x_f|xs) = \log \frac{P^o(x_f|x_s)}{P(x_f|xs)}$$

$$= \log \frac{\Pr(x_f|x_s, o)}{P(x_f|xs)}$$

$$= \log \frac{\Pr(x_s, x_f, o)\Pr(x_s)}{\Pr(x_s, x_f)\Pr(x_s, o)}$$

$$= \log \frac{\Pr(x_s, x_f, o)}{\Pr(x_s, x_f)\Pr(o|x_s)}$$

$$= \log \frac{\Pr(o|x_s, x_f)\Pr(x_s, x_f)}{\Pr(x_s, x_f)\Pr(o|x_s)}$$

$$= \log \frac{p(o|x_s, x_f)}{\mu(o|x_s)}$$

Using this equation, we can rewrite the equation (10) as:

$$\nabla_{\theta_\beta} I(X_f; O|X_s) = \nabla_{\theta_\beta} H(X_f|X_s) - \nabla_{\theta_\beta} H(X_f|X_s, O)$$

$$= \mathbb{E}_{x_s, x, x_f, o}\left[-\nabla_{\theta_\beta}\ell_{\beta^o}(x)\beta^o(x)\Big(\log P(x|x_s) - \log P(x_f|x_s) - \log P^o(x|x_s) + \log P^o(x_f|x_s)\Big)\right]$$

$$= \mathbb{E}_{x_s, x, x_f, o}\left[\nabla_{\theta_\beta}\ell_{\beta^o}(x)\Big(\big(\log P^o(x|x_s) - \log P(x|x_s)\big) - \big(\log P^o(x_f|x_s) - \log P(x_f|x_s)\big)\Big)\right]$$

$$= \mathbb{E}_{x_s, x, x_f, o}\left[\nabla_{\theta_\beta}\ell_{\beta^o}(x)\Big(\log \frac{p(o|x_s, x)}{\mu(o|x_s)} - \log \frac{p(o|x_s, x_f)}{\mu(o|x_s)}\Big)\right]$$

$$= \mathbb{E}_{x_s, x, x_f, o}\left[\nabla_{\theta_\beta}\ell_{\beta^o}(x)\Big(\log p(o|x_s, x) - \log p(o|x_s, x_f)\Big)\right]$$

$$\square$$

## C  IMPLEMENTATION DETAILS

### C.1  THE WHOLE ALGORITHM OF A2IMOC

Algorithm 1 shows a full description of A2IMOC. It follows the architecture of A2C (Mnih et al., 2016; Wu et al., 2017) and has multiple synchronous actors and a single learner. At each optimization step, we update $\pi^o$, $Q_\mathcal{O}$, and $\beta^o$ from online trajectories collected by actors. We update $\hat{p}(o|x_s, x_f)$ for estimating the gradient (11) and $\hat{\mu}(o|x_s)$ for entropy regularization (13). To learn $\hat{p}$ and $\hat{\mu}$ stably, we maintain a replay buffer $B_\mathcal{O}$ that stores option-transitions, implemented by a LIFO queue. Note that using older $o, x_s, x_f$ sampled from the buffer can introduce some bias in the learned $\hat{p}$ and $\hat{\mu}$, since they depend on the current $\pi^o$ and $\beta^o$. However, we found that this is not harmful when the capacity of the replay buffer is reasonably small.

We also add maximization of the entropy of $\beta^o$ to the loss function for preventing the termination probability saturating on zero or one. Then the full objective of $\beta^o$ is written as:

$$\log \hat{p}(o|x_s, x) - \log \hat{p}(o|x_s, x_f) + c_{H_\beta} H(\beta^o(x)),$$

where $c_{H_\beta}$ is a weight of entropy bonus.

### C.2  IMPLEMENTATION OF PPIMOC

For continuous control tasks, we introduce PPIMOC on top of PPO, with the following modifications to A2IMOC.

---

**Algorithm 1** Advantage-Actor InfoMax Option Critic (A2IMOC)

---

1: **Given:** Initial option-value $Q_\mathcal{O}$, option-policy $\pi^o$, and termination function $\beta^o$.
2: Let $B_\mathcal{O}$ be a replay buffer for storing option-transitions.
3: **for** $k = 1, ...$ **do**
4:      **for** $i = 1, 2, ..., N$ **do**                     ▷ Collect experiences from environment
5:          Sample termination variable $b_i$ from $\beta^{o_i}(x_i)$
6:          **if** $b_i = 1$ **then**
7:              Store option transition $x_s, x_f, o_i$ to $B_\mathcal{O}$
8:          **end if**
9:          Choose next option $o_{i+1}$ by $\epsilon$-Greedy
10:          Receive reward $R_i$ and state $x_{i+1}$, taking $a_i \sim \pi_{o_{i+1}}(x_i)$
11:      **end for**
12:      **for all** $x_i$ in the trajectory **do** ▷ Train option-policy, option-values, and termination function
13:          Update $\pi^o(a_i|x_i)$ with PG via the UOAE advantage (12) and policy regularization (13)
14:          Update $Q_\mathcal{O}(x_i, o)$ via the optimistic TD error (12)
15:          **if** $o_i$ has already terminated **then**
16:              Update $\beta^o(x_i)$ via (11) and the maximization of $c_{H_\beta} H(\beta^o(x_i))$
17:          **end if**
18:      **end for**
19:      Train $\hat{p}$ and $\hat{\mu}$ by option-transitions sampled from $B_\mathcal{O}$
20: **end for**

---

**Upgoing Option-Advantage Estimation for GAE**    To use an upgoing option-advantage estimation (12) with Generalized Advantage Estimator (GAE) (Schulman et al., 2015b) common with PPO, we introduce an upgoing *general* option advantage estimation (UGOAE). Letting $\delta$ denote the TD error corresponding to the marginalized option-state-values, $\delta_t = R_t + \gamma V_\mathcal{O}(x_{t+1}) - V_\mathcal{O}(x_t)$, we write the GAE for marginalized policy $\pi_\mu$ as $\hat{A}^o_\mu = \sum_{i=0}^N (\gamma\lambda)^i \delta_{t+i}$, where $\lambda$ is a coefficient. Supposing that $o_t$ terminates at the $t + k$ step and letting $\delta^o$ denote the TD error corresponding to an option-state value $\delta^o_t = R_t + \gamma Q_\mathcal{O}(x_{t+1}, o) - Q_\mathcal{O}(x_t, o)$, we formulate UGOAE by:

$$\hat{A}^o_{\text{UGOAE}} = \begin{cases} \sum_{i=0}^k (\gamma\lambda)^i \delta^o_{t+i} + \max(\underbrace{\sum_{i=k+1}^N (\gamma\lambda)^i \delta_{t+i}}_{\text{upgoing estimation}}, 0) & (k < N) \\ \sum_{i=0}^{N-1} (\gamma\lambda)^i \delta^o_{t+i} + (\gamma\lambda)^N (R_{t+N} + \gamma U_\mathcal{O}(x_{t+N+1}) - Q_\mathcal{O}(x_{t+N}, o)) & (otherwise). \end{cases} \tag{18}$$

The idea is the same as UOAE (12) and is optimistic about the advantage after option termination.

**Clipped $\beta^o$ Loss**    In our preliminary experiments, we found that performing multiple steps of optimization on the gradient (11) led to destructively large updates and resulted in the saturation of $\beta^o$ to zero or one. Hence, to perform PPO-style multiple updates on $\beta^o$, we introduce a clipped loss for $\beta^o$:

$$\nabla_{\theta_\beta} \text{clip}(\ell_{\beta^o}(x) - \ell_{\beta^o_{\text{old}}}(x), -\epsilon_\beta, \epsilon_\beta) \beta^o_{\text{old}}(x) \Big( \log p(o|x_s, x) - \log p(o|x_s, x_f) \Big), \tag{19}$$

where $\epsilon_\beta$ is a small coefficient, $\beta^o_{\text{old}}$ is a $\beta^o$ before the update, and $\text{clip}(x, -\epsilon, \epsilon) = \max(-\epsilon, \min(\epsilon, x))$. Clipping makes the gradient zero when $\beta^o$ is sufficiently different than $\beta^o_{\text{old}}$ and inhibits too large updates.

## D    EXPERIMENTAL DETAILS

### D.1    NETWORK ARCHITECTURE

Figure 8 illustrates the neural network architecture used in our experiments. In Gridworld experiments, we used the same state encoder for all networks and we found that it is effective for diversifying $\pi^o$ as an auxiliary loss (Jaderberg et al., 2017). However, in MuJoCo experiments, we found that sharing

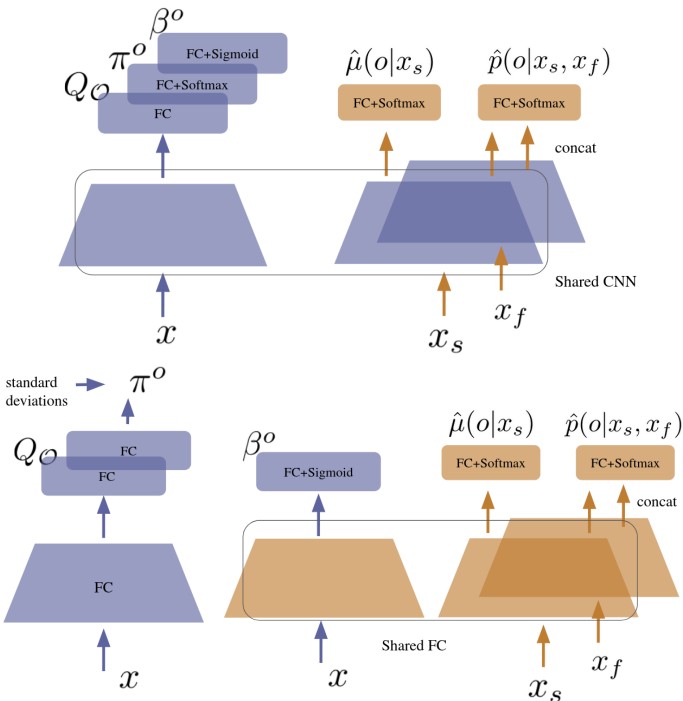

Figure 8: Neural Network architecture used for the Gridworld experiments (top) and the MuJoCo tasks (bottom).

the state encoder can hurt the policy update because the magnitude of $\beta^o$ loss is larger even if clipped loss (19) is used. As a remedy for this, we used two encoders in MuJoCo experiments: one is for $\pi^o$ and $Q_{\mathcal{O}}$, and the other is for $\beta^o$, $\hat{p}$, and $\hat{\mu}$.

In Gridworld experiments, we represented a state as an image and encoded it by a convolutional layer with 16 filters of size $4 \times 4$ with stride 1, followed by a convolutional layer with 16 filters of size $2 \times 2$ with stride 1, followed by a fully connected layer with 128 units. In MuJoCo experiments, we encode the state by two fully connected layers with 64 units. $\pi^o$ is parameterized as a Gaussian distribution with separated networks for standard derivations per option, similar to Schulman et al. (2015a). We used ReLU as an activator for all hidden layers and initialized networks by the orthogonal (Saxe et al., 2014) initialization in all experiments. Unless otherwise noted, we used the default parameters in PyTorch (Paszke et al., 2019) 1.5.0.

## D.2 HYPERPARAMETERS

When evaluating agents, we used $\epsilon$-Greedy for selecting options with $\epsilon_{\text{opt}}$ and did not use deterministic evaluation (i.e., an agent samples actions from $\pi^o$) in all experiments. We show the algorithm-specific hyperparameters of A2IMOC in Table 1. In Gridworld experiments, we used $\epsilon = 0.1$ for AOC. Our AOC implementation is based on the released code[3] and uses truncated $N$-step advantage. Other parameters of AOC and A2C are the same as A2IMOC. We also show the hyperparameters of PPIMOC in Table 2. For PPOC, we used $c_{\mu\text{ent}} = 0.001$ for the weight of the entropy $H(\mu(x))$. Our PPOC implementation is based on the released code[4] and uses $N$-step (not truncated) GAE for computing advantage. PPOC and PPO shares all other parameters with PPIMOC.

---

[3]https://github.com/jeanharb/a2oc_delib
[4]https://github.com/mklissa/PPOC

| Description | Value |
|---|---|
| $\gamma$ | 0.99 |
| Optimizer | RmsProp (Tieleman and Hinton, 2012) |
| RmsProp Learning Rate | $2 \times 10^{-3}$ |
| $\epsilon_{\text{opt}}$ | $0.8 \to 0.2$ |
| Number of timesteps per rollout | 20 |
| Number of actors | 12 |
| $c_{H_\mu}$ | 0.04 |
| $c_H$ | 0.01 |
| $c_{H_\beta} H(\beta^o(x))$ | 0.01 |
| Gradient clipping | 1.0 |
| Capacity of $B_{\mathcal{O}}$ | 480 |
| Batch size for training $\hat{p}$ and $\hat{\mu}$ | 240 |

Table 1: Hyperparameters of A2IMOC in Gridworld experiments.

| Description | Value |
|---|---|
| $\gamma$ | 0.99 |
| Optimizer | Adam (Kingma and Ba, 2015) |
| Adam Learning Rate | $3 \times 10^{-4}$ |
| Adam $\epsilon$ | $1 \times 10^{-4}$ |
| $\epsilon_{\text{opt}}$ | $0.4 \to 0.1$ (Cartpole swingup), $0.2 \to 0.1$ (Otherwise) |
| $\epsilon_\beta$ | 0.05 (Lifelong Billiard),0.1 (otherwise) |
| Number of timesteps per rollout | 256 |
| Number of actors | 16 |
| GAE parameter ($\lambda$) | 0.95 |
| Number of epochs for PPO | 10 |
| Minibatch size for PPO | 1024 |
| $c_{H_\mu}$ | 0.004 |
| $c_\mu$ | 0.001 |
| $c_{H_\beta} H(\beta^o(x))$ | 0.01 |
| Gradient clipping | 0.5 |
| Number of monte carlo rollout to estimate $H(\pi_\mu)$ | 20 |
| Capacity of $B_{\mathcal{O}}$ | 4096 |
| Batch size for training $\hat{p}$ and $\hat{\mu}$ | 2048 |

Table 2: Hyperparameters of PPIMOC in MuJoCo and Classical Control experiments.

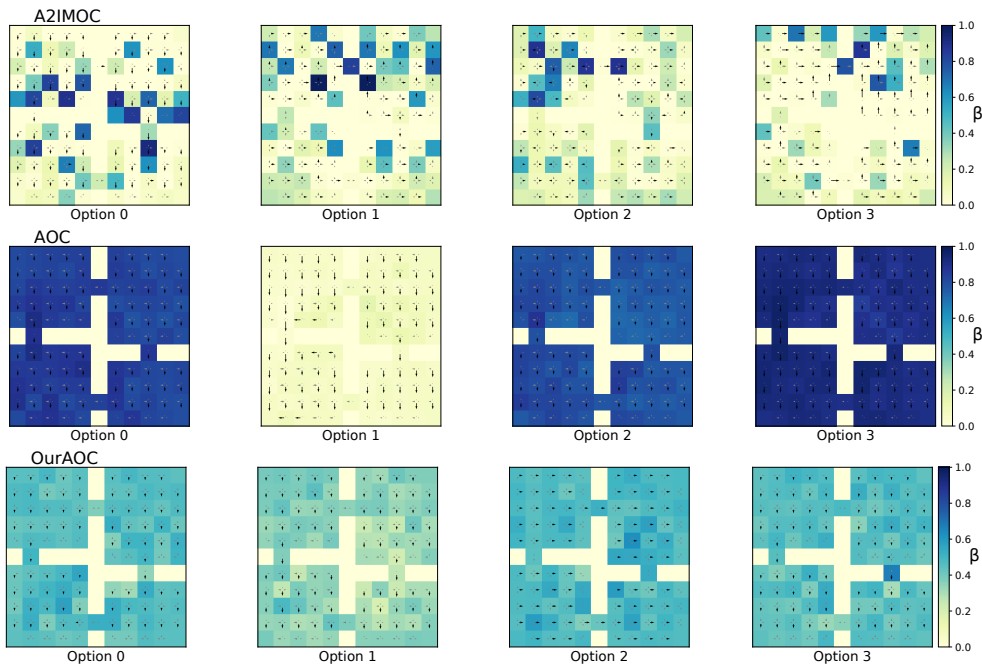

Figure 9: Learned option-policies ($\pi^o$) and termination functions ($\beta^o$) for each option in Gridworld Four Rooms after $5 \times 10^4$ steps. Arrows show the probablities of each action and heatmaps show probabilities of each $\beta^o$. **First row:** A2IMOC. Option 0 tends to go down, option 1 and 2 tend to go right, and option 3 tends to go up. **Second row:** AOC. All options tend to go down. **Third row:** Our AOC. Option 0, 1, and 3 tend to go down and option 2 tends to do right.

### D.3 ENVIRONMENTAL DETAILS

In the Gridworld experiment, an agent can select four actions: go up, go down, go left and go right. With the probability $0.1$, the agent takes a uniformly random action. If the agent reaches one of goals, it receives $+1.0$ or $+2.0$ reward. Otherwise, an action penalty $-0.002$ is given. The maximum episode length is $100$.

MuJoCo Point environments are implemented based on "PointMaze" in rllab (Duan et al., 2016) with some modifications, mainly around collision detection. The maximum episode length is $1000$. In Four Rooms task, an agent receives $+0.5$ or $+1.0$ reward when it reaches a goal. Otherwise, an action penalty $-0.0001$ is given. This reward structure is the same in Billiard Task: an agent receives a goal reward when the object ball reaches a goal, otherwise it receives penalty.

### D.4 EARLY OPTION-POLICIES LEARNED IN GRIDWORLD FOUR ROOMS

Figure 9 shows early option-polices and termination probabilities in Gridworld Four Rooms experiment. We can see that A2IMOC learned the most diverse option-policies.

### D.5 QUALITATIVE ANALYSIS OF POINT FOUR ROOMS EXPERIMENT

Figure 10 shows the visualizations of learned option-polices and termination functions in MuJoCo Point Four Rooms, averaged over $100$ uniformly sampled states per each position. Arrows show the expected moving directions, computed from rotations and option-policies. Terminating regions and option-policies learned with PPIMOC are diverse. For example, option 1 tends to go down while option 2 tends to go right. In the sampled trajectory of PPIMOC, we can see that it mainly used option 1 but occasionally switched to option 0 and option 2 for reaching the goal, and switched to option 3 around the goal. Contrary, for reaching the goal PPOC only used option 3 that does not terminate in any region. Options learned by Our PPOC is almost the same: termination probability is high around the upper left corner and option-policies direct downward.

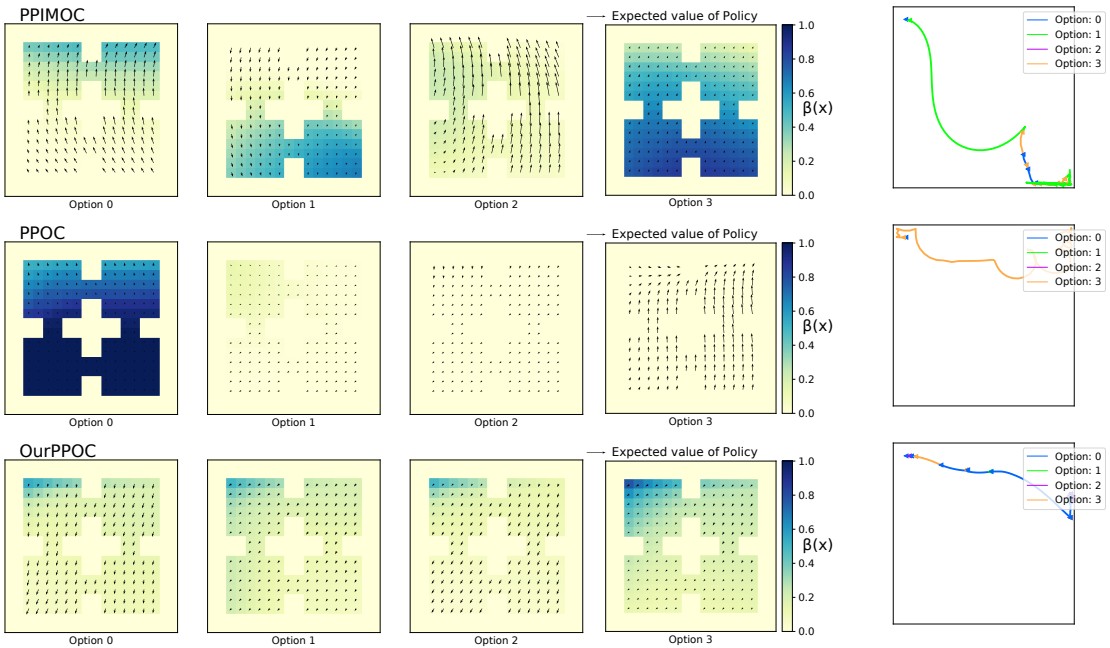

Figure 10: **Left:** Learned option-policies ($\pi^o$) and termination functions ($\beta^o$) in MuJoCo Point Four Rooms experiment. Arrows show the expected moving direction of the agent and heatmaps show probabilities of each $\beta^o$. **Right:** Sampled trajectories of each method. **First row:** PPIMOC. Terminattion regions are clearly separated and option-polices are diverse. **Second row:** PPOC. Option 0 terminates at almost everywhere, while option2 and 3 does not terminate anywhere. **Third row:** Our PPOC. All options are almost the same.

### D.6 ABLATION STUDIES

We conducted ablation studies with three variants of A2IMOC/PPIMOC:

- $c_{H_\mu} = 0$: Do not use the policy regularization based on MI (13).
- $N$-step Advantange: Use $N$-step advantage or $N$-step GAE instead of UOAE (12) UGOAE (18).
- Truncated $N$-step Advantage: Compute advantage ignoring future rewards instead of using UOAE or UGOAE.

Figure 11 shows all results in three tasks. We can see that UOAE is effective in all tasks, since both $N$-step advantage and truncated $N$-step advantage performed worse than UOAE. The policy regularization based on MI (13) is effective only in the Point Billiard lifelong learning task.

### D.7 NUMBER OF OPTIONS

Figure 12 shows the performance of IMOC with varying the number of options. Two options perfomed worse in all experiments and we need four or more options to make use of IMOC. However, when we increase the number of options to six and eight, we don't see any peroformance improvement from four options, despite of our analysis that we need sufficiently many options to cover the state space Appendix A. This would be an interesting problem for future works.

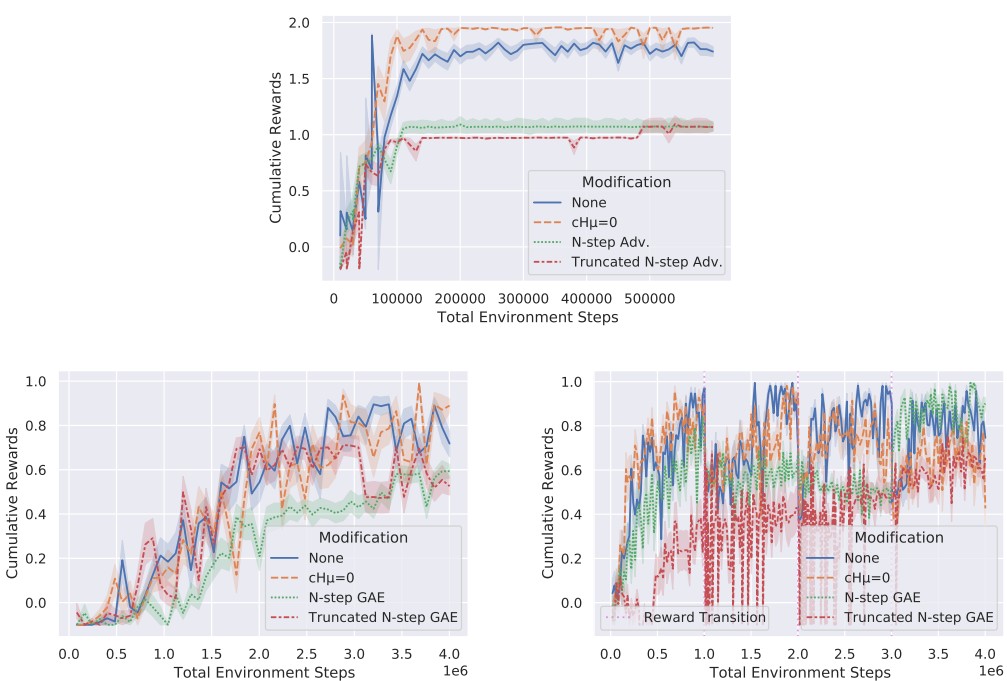

Figure 11: Ablation studies. **Top:** Peformance progression of A2IMOC in Gridworld Four Rooms. **Bottom:** Peformance progression of PPIMOC in MuJoCo Point Four Rooms (left) and MuJoCo Point Billiard (right).

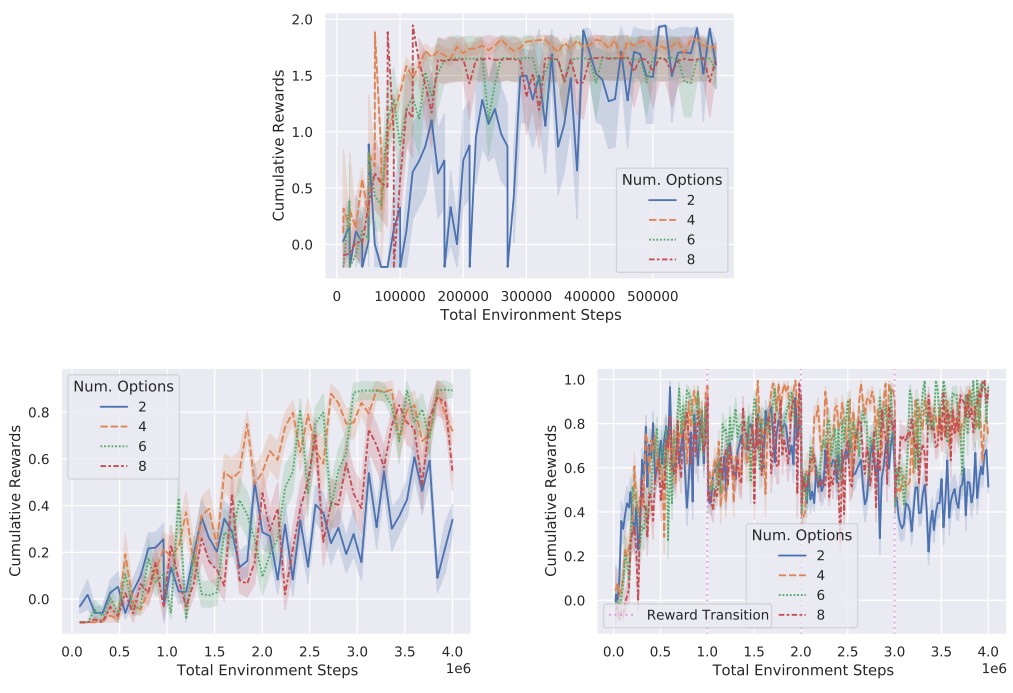

Figure 12: Ablation studies. **Top:** Peformance progression of A2IMOC in Gridworld Four Rooms. **Bottom:** Peformance progression of PPIMOC in MuJoCo Point Four Rooms (left) and MuJoCo Point Billiard (right).

