# OpenReview forum: "Diverse Exploration via InfoMax Options"
_ICLR.cc/2021/Conference — Reject_

### Official Review · AnonReviewer1 · 2020-10-29
**Official Blind Review**

**Rating:** 5
**Confidence:** 5

**Review:**

This is an interesting paper that investigates the use of options for improved exploration by encouraging diverse termination functions via a mutual information measure. The paper is well written and mostly clear (with a few points below):

Overall I think the idea has merit, but I think the experiments fall short of demonstrating its performance. What seems to be lacking to me is:
1. Demonstrating performance with stochastic transitions (I believe all experiments are with deterministic transitions).
2. Overcoming the coarse approximation for the infomax objective (7) (see point 5 below).
3. Clarifying the details of the classification model, which seems central to the algorithm (see point 2 below).

Some questions for the authros:
1. What is the law of $X_S$ in equation (7)? In particular, if $X_f|X_s$ is the random variable denoting a state transition, wouldn't this just be the stationary distribution of the current policy? Further, in Proposition 1, the first state $x_s$ isn't sampled (at least not by $\pi^o$ nor $\beta^o$), so it's not clear what the random variable $X_S$ represents.
2. Right before section 4, the authors write "learning a classification model over options $\hat{p}(o|x_s, x_f)$ from sampled option transitions". This seems central to the algorithmic performance, but it's not clear how it gets used.
3. In equation (12), in the condition for $b_t = 1$, it seems it's saying the same option ($o_t$) should be picked again. Is this the case?
4. Above equation (13), the authors write "Supposing that the current option ot terminates at the $t + k$ step". How is this calculated? This seems especially problematic with stochastic transitions.
5. In page 5 the authors write "[$I(A; O | X_S)$] can be interpreted as a local approximation of the infomax objective (7), assuming that each action leads to different terminating regions." This seems like a rather rough approximation; in particular, doesn't this simply encourage diversity of one-step action transitions as opposed to option termination?
6. In Figure 1b the difference between A2IMOC and A2C does not appear to be statistically significant (there is major overlap of the confidence regions). Does this not suggest merely marginal gains?
7. In Figure 2, were all algorithms using 4 options?
8. Were any of the environments evaluated with stochastic transitions?
9. How many options were used in the MuJoCo environments?
10. What do the "sample" indicators mean in your proof of Lemma 1? This suggests the proof is mostly a modification of the proof in Harutyunyan et al. (2019).
11. Do you have a similar plot as Figure 8, but for the MuJoCo experiments?

Minor issues:
1. Given that you're dealing in episodic tasks, below equation (1), shouldn't the $Q^{\pi}$ values be indexed by timestep?
2. In the definition of $V_{\mathcal{O}}(x)$, it should be indexed by $\mu$.
3. Right above section 2.2, remove the extraneous "and".
4. In equation (4), $Q_{\mathcal{O}}$ is missing the second parameter.
5. You should state equation (6) as a proper theorem.
6. Equation (10) is over the margins.
7. In the first sentence of the paragraph above equation (14), remove the extraneous "common"
8. In the section titled **MuJoCo Point Four Rooms** there is a typo, it says "Figure 3b" but it should say "Figure 3a".
9. In Related Work, under **Options for Exploration** you could also cite some related work:
  - "Using bisimulation for policy transfer in MDPs", Castro & Precup, AAAI 2010
  - "Automatic construction of temporally extended actions for mdps using bisimulation metrics", Castro & Precup, EWRL, 2011

---

> ### Author Response · Authors · 2020-11-20
> **Response to Reviewer 1**
>
> Thank you for your kind and constructive reviews.
>
> Here are our answer to the concerns:
>
> 1. As we describe below, Gridwolrd is stochastic. We are happy to add experiments in some more interesting stochastic environments, though. Also, we share the concern with stochasticity since $\hat{p}$ depends on both policy and environmental dynamics. Some empirical analysis would be interesting.
>
> 2. The main contribution of this paper is the method for learning diverse termination conditions. We agree that the policy regularization via maximizing $I(A;O|X_s)$ is a relatively coarse approximation of the targeting MI $I(X_f; O|X_s)$. However, we need pseudo rewards to maximize the MI via the policy parameters. Thus, to keep the paper simple, we used this rough but simple regularization.
> 3. We had described the training details in Appendix B.2. We are happy to add some more explanations.
>
> Here are our answer to the questions:
>
> 1. $X_s$ denotes a starting state of an option. $\Pr(X_s = x)$ is the probability that an agent starts to use a new option at a state $x$. Thus, $X_f|X_s$ is the random variable denoting a state transition accompanied by an option. We will make the context clear in the revised paper.
> 2. We had described how we estimated the $\hat{p}$ in Appendix B.1. Also, we are preparing a further analysis for $\hat{p}$.
> 3. We found that it was our mistake. Thank you for pointing it out. An agent chooses the same option when $b_t = 0$.
> 4. $k$ is not a constant but depends on the sampled termination variables $b_t, b_{t + 1}, …$, collected along the $N$-step rollout. Precisely, we define the $k$ so that $b_t = b_{t + 1} = … = b_{t + k - 1} = 0$ and $b_{t + k} = 1$ hold. We will clarify this in the revised version.
> 5. We agree that this is a rough approximation. Since our main contribution is in training $\beta$, we used this regularization for simplicity.
> 6. We agree that the gain looks marginal. IMOC is somehow unstable when evaluating and occasionally fails (= got no positive reward). We still don't have a remedy for this instability.
> 7. Yes, all algorithms used four options.
> 8. Yes. As we had noted in Appendix C.3., the Four Rooms Gridworld is stochastic in that each action can fail with the probability of 0.1, resulting in a uniformly random transition. We will note this more clearly.
> 9. Four options are used in MuJoCo experiments as well as other experiments.
> 10. The sample indicators mean that the corresponding terms are replaced by expectations in equations 17 and 18. We will improve our paper to be more clear. These indicators are actually borrowed from Harutyunyan et al. (2019) to help understand the equations.
> 11. Figure 8 is for MuJoCo experiments. We are happy to add similar figures for Gridworld or Billiard experiments.
>
> Also, thank you for pointing out lots of minor issues.
>
> > Given that you're dealing in episodic tasks, below equation (1), shouldn't the $Q^{\pi}$ values be indexed by timestep?
>
> > In the definition of $V_{\mathcal{O}}(x)$, it should be indexed by $\mu$.
>
> We agree that indexing makes the notations clearer. However, to keep them compatible with Bacon et al. (2017), we are to leave them as are.

---

> > ### Comment · AnonReviewer1 · 2020-11-20
> > **Highlight changes?**
> >
> > Thank you for your response. Would it be possible to highlight the changes made to your draft to facilitate my revision of it?

---

> > > ### Author Response · Authors · 2020-11-21
> > > **We have not yet uploaded the revised paper**
> > >
> > > We are sorry but still working on the first revision. We will post a general comment when we upload it.

---

### Official Review · AnonReviewer2 · 2020-10-29
**Very nice idea, but unconvincing results**

**Rating:** 4
**Confidence:** 4

**Review:**

This paper seeks to learn diverse options for exploration,
principaling by combining two techniques: the termination gradient
theorem (Harutyunyan et al. 2019) and empowerment (in the spirit of
VIC/DIAYN). This is an appealing idea, and the approaches to
empowerment (such as for DIAYN) are a great fit for the limitations of
the Harutyunyan work.

However, there were a number of issues I found with the paper which
currently leaves me leaning more towards rejecting in the current
form. Again, solid idea, but both the paper itself and the
experimental results (thus the method) need further work.

Comments/Questions:

- Just above Prop 2: The use of the discriminator / classification
model here should be attributed to related work.

- Uncertainty-aware option selection: How is this like UCB? This is
not 'uncertainty', it is the stochasticity of the policy, so this
actually just looks more like entropy regularization as it is
frequently used. Am I missing something?

- Section 5, Gridworld:  Do episodes end as soon as the first non-zero
reward is found? I don't think that this is a particularly good choice
of domain for studying exploration.

- "A2IMOC performed the best and found the optimal goal in most trials."
I don't see any evidence for either part of this statement. Judging by
Figure 1, none of the methods are reliably achieving the 2.0 goal. And
judging by the error bars, it is difficult to say that A2IMOC is
better than AOC. I'd add that looking at Figure2, the policies learned
by the different options for A2IMOC look to terminate in different
states, but appear to have essentially the same behavior policy. This
looks like one of the common failure cases of OC, where the options
either learn the primitive actions (e.g. Figure 2, AOC) or all learn
identical policies (e.g. Figure 2, A2IMOC).

As with Figure 1, the learning curves in Figure 2 do not allow us to
draw any real conclusions about PPIMOC being an improvement over PPO.
Though, Figure 3, does seem to show some significant improvement in
this non-stationary task setting. But this is perhaps the only
positive empirical result, and thus is a lot to hang the paper on.

- Appendix: epsilon-greedy ablation looks to be actually performing
the best, why not use this instead of the UAOS?

Pros:

Interesting idea, very appealing intuitively
Small, but highly relevant, theoretical contributions

Cons:

Experimental results are wholly unconvincing
Uncertainy-aware option selection needs to be reframed
Writing could use some work, especially around putting this in the
proper context of existing work.

---

> ### Author Response · Authors · 2020-11-20
> **Response to Reviewer 2**
>
> Thank you for your kind and constructive reviews.
>
> > Uncertainty-aware option selection: How is this like UCB? This is not 'uncertainty', it is the stochasticity of the policy, so this actually just looks more like entropy regularization as it is frequently used. Am I missing something?
>
> We agree that it's just like entropy regularization and not provided with proper explanations. We are happy to revise the whole subsection.
>
> > Section 5, Gridworld: Do episodes end as soon as the first non-zero reward is found?
>
> Yes, episodes terminate when an agent reaches one of the goals.
>
> > I don't think that this is a particularly good choice of domain for studying exploration.
>
> We agree that our problem set does not cover some interesting hard-exploration problems (e.g., single-goal but sparse reward environments such as Montezuma's Revenge). On the other hand, we argue that IMOC, in the current form, does not generally improve the regret bound but works effectively as an inductive bias in problems with certain properties: e.g., environments with suboptimal goals. We are happy to add some additional experiments, but we guess that the current IMOC can address a limited type of problem.
>
> > I'd add that looking at Figure2, the policies learned by the different options for A2IMOC look to terminate in different states, but appear to have essentially the same behavior policy.
>
> This is due to the strong 2.0 reward signal. The current IMOC doesn't use pseudo rewards, thus sensitive to the reward scale of an environment. In the experiments, we observed that the option-policies of IMOC got closer as the training progressed. We hope to add additional figures to support this claim in the revised paper.
>
> > especially around putting this in the proper context of existing work.
>
> We found that our explanation of the relationship between IMOC and other methods with state-conditioned option probability models (e.g., VIC and DIAYN) was insufficient. Also, we gratefully appreciate some more suggestions if you have any in your mind.

---

### Official Review · AnonReviewer3 · 2020-10-29

**Rating:** 5
**Confidence:** 3

**Review:**

Review for "Diverse Exploration via Infomax Options"

Thank you for your feedback. I found that the analysis of the method in the revision is informative. However, the comparison with the baselines is still lacking, and the experiments are only performed in simple environments. For these reasons, I keep my rating unchanged after the rebuttal.

=============

Summary:
This paper studies the problem of discovering options for exploration in reinforcement learning. To this end, they propose infomax criteria that maximize the information between options and option-terminating states conditioned on the option-starting states. The paper derives update rules for the infomax criteria and applies the gradient descent algorithm to optimize options. The proposed method is evaluated at Gridworld/Mujoco four rooms tasks and Point Billiard environment, and the authors show that the resulting options are diverse.


Pros:
- The information maximization criteria for diverse option-terminating states is novel.
- The paper does a nice job describing the required background materials in great detail.
- The paper clearly describes the objective and the corresponding update rule with implementation details.


Cons:
- The experimental results are too weak to prove the effectiveness of the proposed method. In detail, the paper's experiment tasks are too simple, e.g., all tasks' goals are to reach one of the four points in a simple square room. Moreover, from Figure 3-(b), it doesn't seem that the proposed method (PPIMOC) outperforms the baselines.
- The performance comparison with important baselines such as termination critic (TC) is absent in the experiments.
- The paper suggests some techniques such as uncertainty-aware option selection or upgoing option-advantage estimation, but the paper lacks ablation studies.
- It is not clear that maximizing the mutual information in Equation (7) leads to diverse options. The paper should provide theoretical results or more experimental analysis to make the case.


Comments:
- Experiments on more challenging tasks, such as Montezuma's revenge from Atari games, could improve the claim of the paper.
- I recommend performing an ablation study of the proposed algorithms (in Section 4) to isolate the effect of each technique.
- I am not fully convinced that information maximization leads to diverse options. Aren't there any problems associated with the collapse of options or terminating states?

(minor)
There are some typos.
- End of Section 2.1: "with and o and not"
- Equation (4) and the following sentence: Q_O(x) -> Q_O(x,o)
- Last paragraph of Section 2.3: "Contrary to the terminatio"


In summary, the paper suggests a new perspective on diverse options via infomax criteria, but the experimental results are too weak, and the analysis of the method is lacking.

---

> ### Author Response · Authors · 2020-11-20
> **Response to Reviewer 3**
>
> Thank you for your kind and constructive reviews.
>
> > The performance comparison with important baselines such as termination critic (TC) is absent in the experiments.
>
> We implemented TC but failed to reproduce the results in their paper. Thus, we excluded our TC implementation from the comparison for fairness.
>
> > The paper suggests some techniques such as uncertainty-aware option selection or upgoing option-advantage estimation, but the paper lacks ablation studies.
>
> We had conducted ablation studies in Appendix C.5. Please let us know if you suggest more experiments.
>
>
> > Experiments on more challenging tasks, such as Montezuma's revenge from Atari games, could improve the claim of the paper.
>
>  We are happy to add some additional experiments, although we don't expect that IMOC improves performance for single-goal problems. We interpret IMOC as an inductive bias for visiting diverse regions, enhancing exploration in limited tasks.
>
> > I am not fully convinced that information maximization leads to diverse options. Aren't there any problems associated with the collapse of options or terminating states?
>
> Thank you for pointing this out. Although we need more analysis, we believe that this MI maximization produces diverse options given trajectory samples $(x_s, x_f, o)$ many enough to cover the whole state space. Thus, the most plausible source of failure is that we possibly have too few trajectories, ironically meaning that we need to explore the state space to obtain explorative options. We hope to add a detailed analysis in the revised paper about both the MI maximization itself and the quality of sample-based estimation we employed.

---

### Official Review · AnonReviewer4 · 2020-11-09
**The paper proposes a variant of the option critic algorithm for hierarchical reinforcement learning. It would benefit from an analysis of the behaviour of the algorithm in a broader range of problems.**

**Rating:** 4
**Confidence:** 4

**Review:**

The authors propose a modification of the option-critic algorithm for hierarchical reinforcement learning. The proposed algorithm modifies how the termination conditions of the options are improved by experience. Specifically, the algorithm aims to maximize the mutual information between the options and their termination states. The authors develop an optimization scheme for achieving this objective and provide empirical results in a number of domains

The empirical results help understand the scalability of the approach but they are less useful in evaluating how useful the proposed algorithm is in general. The experimental domains used in the paper are not as diverse as one would wish to see. In addition, across domains, the general structure of the task is the same: there are multiple goal regions offering different amounts of reward; the challenge for the agent is to not get distracted by the lower offerings. So it would be informative to see the algorithm analysed in a more varied set of problems, including problems with a single goal region. Also informative would be to execute the algorithm with varying numbers of options and examine the effect.

The logic behind using uncertainty-aware option selection is not clear. Further discussion and exploration of it in the paper would be useful.

Bottom of page 4: “By adding the uncertainty bonus, this strategy encourages selecting lesser-known options.” Given  that $\mu(o, s_t)$ is the probability of choosing option $o$ in state $s_t$, which is not directly related to how well known an option is, it is not clear to me how Equation 12 encourages selecting “lesser-known” options.

Bottom of page 5: “Thus, an agent is required to switch its behavior around a suboptimal goal (e.g., from “going down” to “going right”), which options can help.” This needs clarification. How do options help with this? And why is that not achievable with primitive actions?

In Figure 2, OurAOC lags behind AOC. This could be noted in the paper and the reasons can be discussed.

Figures 2 and 6 are very useful.

Section 2.3,  perfoms --> performs
Section 2.3,  terminatio --> termination

Author feedback: The behaviour and performance of the algorithm in single-goal environments should be part of the paper. The paper should show not only where the algorithm succeeds but also where it fails. I appreciated the author's efforts to add diversity to the domains evaluated but they do not go far enough to change my score.

---

> ### Author Response · Authors · 2020-11-20
> **Response to Reviewer 4**
>
> Thank you for your kind and constructive reviews.
>
> >  In addition, across domains, the general structure of the task is the same: there are multiple goal regions offering different amounts of reward; the challenge for the agent is to not get distracted by the lower offerings.
>
> We agree that they share the same structure and we are happy to add some more experiments. However, we note that IMOC didn't perform well in single-goal environments in our preliminary experiments. IMOC tried to visit diverse regions in such environments even after finding the goal, resulting in the slow convergence.
>
> > Thus, an agent is required to switch its behavior around a suboptimal goal (e.g., from "going down" to "going right"), which options can help." This needs clarification.
>
> In our experiments, we observed that RL methods without options failed to learn multimodal policies that can lead the agent to multiple goals, converging to sub-optimal policies. Contrary, diverse options encourage multimodal behaviors and prevent agents from converging to a suboptimal policy.
>
> > The logic behind using uncertainty-aware option selection is not clear.
>
> We agree that this is provided with improper explanations and should be updated in the revised version.

---

### Author Response · Authors · 2020-11-24
**Revision Summary**

Thank you for all your constructive and thoughtful reviews.
We summarize the changes in the revision as follows:
1. Add a simple analysis of deterministic InfoMax options in a deterministic chain environment.
2. Remove the Uncertainty-Aware Option Selection. Use ε-Greedy in all experiments.
3. Add some additional experiments (MountainCar, Cartpole swing up).
4. Describe the relationship between our work and [1] more.

Since the time constraints, we could not address all concerns, mostly i) the weakness of experimental results and ii) lack of experiments of a diverse set of problems. However, we believe that our responses and the revision address many of your concerns. We are happy to provide some more clarifications.

[1]: K. Gregor, D. J. Rezende, and D. Wierstra. Variational intrinsic control. In 5th International
Conference on Learning Representations, ICLR 2017, Toulon, France, April 24-26, 2017, Workshop
Track Proceedings. OpenReview.net, 2017. URL https://openreview.net/forum?id=Skc-Fo4Yg.

---

> ### Comment · AnonReviewer1 · 2020-11-24
> **Highlight changes in paper?**
>
> Hi, it does not seem like your changes are highlighted in the newest revision? As I suggested below, it'd be great if you could use color to clarify what changed from the original submission.

---

> > ### Author Response · Authors · 2020-11-25
> > **Maybe the system will automatically run pdfdiff?**
> >
> > Although we are not familiar with the review process of ICLR and not sure, the [CFP](https://iclr.cc/Conferences/2021/CallForPapers) says:
> > > a pdfdiff will be done against the submission at the paper submission deadline.
> >
> > So we guess that the system will run pdfdiff automatically.
> >
> > Also, we can view the diff by clicking `Show Revisions` and then `Compare Revisions`. Maybe [this link](https://openreview.net/revisions/compare?id=OtAnbr1OQAW&left=QbEmIy-trw&right=tgMSzz3Ilq) would work (the right one is newer).

---

### Decision · Program_Chairs · 2021-01-07
**Final Decision**

**Decision:**

Reject

**Comment:**

The paper introduces a variant to the option-critic framework that encourages options to display a certain level of "diversity" and this is induced by introducing a mutual-information objective between the options and their transitions. The authors conjecture that such criterion makes options more suitable for exploration.

Overall, reviewers agree that the idea behind the proposed method and the general approach is sound and interesting. Nonetheless, there is general consensus that the current submission suffers from a number of shortcomings that make it unsuitable for acceptance.

Following the detailed comments provided by the reviewers, I strongly encourage the authors to focus on the following dimensions to improve the paper:
1- The current experiments indeed provide a first illustration of how the proposed algorithm works, but they need significant improvement in variety and scope: As pointed out by the reviewers, the current experiments do not cover single-reward challenging exploration benchmarks (such as Montezuma). I agree with the authors that the inductive bias implemented in their algorithm is designed with diversity of goals in mind, but if the main point is to improve exploration, it is natural to expect results in that respect. Alternatively, the authors should state more explicitly the type of problems their method is intended to solve from the very beginning of the paper and design experiments accordingly.
2- The initial mutual information objective is simplified across multiple steps and it is unclear how much the approximations impact the original "semantic" of the objective.
3- A more thorough comparison with mutual-information-based methods such as DIAYN or VIC is needed. Also, I wonder what is the connection with more goal-based exploration approaches such as GoExplore or SkewFit.